# A Competition Winning Deep Reinforcement Learning Agent in microRTS

## Abstract

Scripted agents have predominantly won the five previous iterations of the IEEE microRTS ($\mu$RTS) competitions hosted at CIG and CoG. Despite Deep Reinforcement Learning (DRL) algorithms making significant strides in real-time strategy (RTS) games, their adoption in this primarily academic competition has been limited due to the considerable training resources required and the complexity inherent in creating and debugging such agents. AnonymizedAI is the first DRL agent to win the IEEE microRTS competition. In a benchmark without performance constraints, AnonymizedAI regularly defeated the two prior competition winners. This first competition-winning DRL submission can be a benchmark for future microRTS competitions and a starting point for future DRL research. Iteratively fine-tuning the base policy and transfer learning to specific maps were critical to AnonymizedAI's winning performance. These strategies can be used in economically training future DRL agents. Further work in Imitation Learning using Behavior Cloning and fine-tuning these models with DRL has proven promising as an efficient way to bootstrap models with demonstrated, competitive behaviors.

## 1 Introduction

Deep reinforcement learning (DRL) has proven to be powerful at solving complex problems requiring several steps to achieve a goal, such as Atari games (Mnih et al., 2013), continuous control tasks (Lillicrap et al., 2016), and even real-time strategy (RTS) games like StarCraft II by AlphaStar (Vinyals et al., 2019). However, AlphaStar was trained with thousands of CPUs and GPUs/TPUs for several weeks. RTS games are particularly challenging for DRL for several reasons: (1) the observation and action spaces are large and varied with different terrain and unit types; (2) each unit type can have different actions and abilities; (3) each action can control several units at once; (4) rewards are sparse (win, loss, or tie) and delayed by possibly several thousand timesteps; (5) winning requires combining tactical (micro) and strategic (macro) decisions; (6) actions must be taken in real-time (i.e., the game won't wait for the agent to take an action); (7) the agent might not have full visibility of the game state (i.e., fog of war); and (8) events in the game might be non-deterministic.

microRTS (stylized as $\mu$RTS) is a minimalist, open-source, two-player, zero-sum RTS game testbed designed for research purposes (Ontañón, 2013). It includes many aspects of RTS games, simplified: different unit types, unit-specific actions, terrain, resource collection and utilization to build units, and unit-to-unit combat where units have different strengths and weaknesses. microRTS also supports fog of war and non-determinism; however, these were disabled for the IEEE-CoG2023 microRTS competition.

The IEEE microRTS competitions have been hosted at the Conference on Games (CoG) nearly every year since 2019 and at the Conference on Computational Intelligence and Games (CIG) before that since 2017 (Ontañón et al., 2018). Competitors submit an agent that plays against other submissions and baselines in a round-robin tournament on 12 different maps: 8 Open (known beforehand) and 4 Hidden (unknown until after the competition results are released). Agents are supposed to submit actions every step within 100 ms. Without GPU acceleration, this is a significant constraint for deep neural network agents.

This paper describes how the AnonymizedAI agent[1] was trained and became the first DRL agent to win the microRTS competition by winning at CoG in 2023. The agent consists of 7 trained neural networks, taking 70 GPU-days to train. This significant training time combined with the general difficulty in debugging and fine-tuning a DRL implementation could explain why DRL hasn't been competitive so far. However, we demonstrate that transfer learning to specific maps was critical to winning the competition. This strategy and our training framework can be a starting point for future research and competition agents.

While microRTS doesn't support human players, the competitions have made several agents available to use for imitation learning. Work following the competition shows that behavior cloning and fine-tuning with DRL can be used to train a competitive agent more economically. Using the same playthroughs to train the critic heads on win-loss rewards means that DRL can be trained with just sparse win-loss rewards, eliminating the need for a handcrafted reward function.

## 2 RELATED WORK

### 2.1 MICRORTS-PY

Huang et al. (2021) released MicroRTS-Py[2], an OpenAI Gym wrapper for microRTS that includes a Proximal Policy Optimization (PPO) (Schulman et al., 2017) implementation trained on 1 of the Open maps (`16x16basesWorkers`). They added action composition, a shaped reward function, invalid action masking, IMPALA-CNN (a convolutional neural network with residual blocks), and trained against a diverse set of scripted agents. The agent achieved a 91% win rate on a single map against a diverse set of competition bots.

In their ablation studies, they found invalid action masking was essential to have an agent that could compete at the most basic level (82% win rate with invalid action masking, 0% without). Using the residual block network IMPALA-CNN architecture instead of the Atari Nature CNN by Mnih et al. (2013) got the win rate up the rest of the way.

They experimented with two different ways to issue player actions: Unit Action Simulation (UAS) and GridNet (Han et al., 2019). UAS calls the policy iteratively on each unit, simulating the game state after each unit action before issuing all unit actions combined to the game engine. GridNet computes the actions for all units in a single policy call by computing unit action logits for all grid positions and using a player action mask to ignore cells that don't have any units owned by the player. UAS performed better than GridNet (91% vs 89%). Despite UAS's better performance, MicroRTS-Py is deprecating UAS in favor of GridNet because of UAS's more complex implementation and difficulty to incorporate self-play and imitation learning, both features important in AnonymizedAI and our further work.

They tried training with self-play, where the policy plays against itself. They found self-play didn't improve the win rate, neither when only using self-play nor when training with half self-play and half scripted bots. We found a bug where resources (which should be unowned) were being counted as owned by the opponent if the agent was the second player, which likely contributed to their finding no improvement.

We reimplement much of MicroRTS-Py and extend its capabilities to support training on more maps, extend training capabilities, fix self-play[3], and add imitation learning.

### 2.2 DEEPMIND ALPHASTAR

Vinyals et al. (2019)'s AlphaStar is a grandmaster-level AI trained with DRL to play the RTS game StarCraft II. They created an initial set of agents through imitation learning: supervised learning using a dataset of observations, actions, and rewards to train a policy to mimic actions from the dataset. The dataset was created by sampling replays of top-quartile human players. The supervised agents were rated in the top 16% of human players and used as starting points for DRL. They created a league-based framework to train multiple agents in parallel, each with different opponents to beat,

---

[1] AnonymizedAI's microRTS GitHub URL

[2] https://github.com/Farama-Foundation/MicroRTS-Py

[3] Anonymized GitHub URL to opponent owned resources bug fix

thus creating a diverse set of training agents. AlphaStar was trained on 3072 TPU cores and 50,400 preemptible CPU cores for a duration of 44 days.

microRTS is a much simpler game than StarCraft II, both in game mechanics and simulation cost. Huang et al. (2021) trained on a single map for 300 million steps in less than 3 GPU-days. AnonymizedAI trained on 10 maps for 1.5 billion steps in 70 GPU-days. While AnonymizedAI didn't use supervised learning to bootstrap the agent, our following work uses imitation learning to train a competitive agent.

AlphaStar's observation and action space is significantly different from microRTS. The StarCraft II Learning Environment (PySC2) is made to be similar to a human player's observations and controls. An AlphaStar action is (1) selecting an action type, (2) selecting a subset of units to perform the action on, and (3) selecting a target for the action (either a map location or visible unit). Once supplied an action and a target, units will perform the action until the action is complete or the unit is interrupted. microRTS requires the agent to give single step actions for each unit at each timestep.

### 2.3 LUX AI KAGGLE COMPETITIONS

The competition platform Kaggle hosts Simulation competitions where competitors submit agents that play against other submitted agents in a turn-based game environment. About once a year since 2020, Kaggle features an RTS-like Simulation competition: Halite, Lux AI, Kore, and Lux AI Season 2. Rules-based agents won Halite, Kore, and Lux AI Season 2. The first season Lux AI winning DRL agent by Pressman et al. (2021) has many similarities to MicroRTS-Py: GridNet action space, reward shaping, and an actor-critic training algorithm (IMPALA with additional UPGO and TD($\lambda$) loss terms, instead of PPO). Instead of training with a shaped reward function throughout training, Pressman et al. (2021) used shaped rewards on a smaller map before transitioning to sparse win-loss rewards on larger and competition-size maps. The top DRL agent by Limburg (2023) in the Lux AI Season 2 competition used a "DoubleCone" neural network backbone with critic and actor heads. DoubleCone is similar to ResNet's backbone but the middle residual blocks are downscaled 4x to reduce inference time. AnonymizedAI switches from shaped to sparse rewards during training and uses the DoubleCone architecture.

## 3 METHODS

### 3.1 NEURAL NETWORK ARCHITECTURE

We reimplement much of MicroRTS-Py, including the PPO implementation, action composition, shaped reward function, invalid action masking, GridNet, self-play, and scripted bot training. AnonymizedAI's codebase[4] supports multiple environments and reimplementing allowed the environment to fit into the existing codebase. We extended the observation representation in two ways: (1) walls and (2) unit destinations as invalid move targets in the invalid action mask. Only the unit's current location was considered invalid in MicroRTS-Py; however, microRTS doesn't allow units to move where another unit is moving into. Padded positions were represented as walls, which are impassable and noninteractive.

AnonymizedAI loads 7 different policy networks. Only one network is used for a given map (Table 5). Networks are selected by (1) gathering all networks that are compatible with this map and its size, (2) prioritizing map-specific networks over size-specific networks, and (3) picking the highest priority network that can run within the allotted time on the current hardware. Multiple networks can be prioritized for the same map or map size. microRTS supports any map size (even non-square), and observations are padded to fit the policy network. Policy actions are clipped to fit the map size.

AnonymizedAI uses two different neural network backbones: Limburg (2023)'s DoubleCone(4, 6, 4) and a custom network squnet. The actor head is a convolutional layer that outputs logits for unit actions at every position. A unit action is composed of independent discrete subactions: $D = \{a_{action\ type}, a_{move\ direction}, a_{harvest\ direction}, a_{return\ direction}, a_{produce\ direction}, a_{produce\ type}, a_{relative\ attack\ position}\}$. Invalid action masking sets logits to a very large negative number (thus zeroing probabilities and gradients) for actions that are illegal or would accomplish

---

[4]AnonymizedAI's GitHub URL

nothing (e.g., moving a unit to an occupied or reserved position) (Huang & Ontañón, 2020). This masking significantly reduces the action space per turn and makes training more efficient.

While DoubleCone could support any map size, inference would likely exceed 100 milliseconds for larger maps. Therefore, for larger maps, we used a different architecture, squnet, which nests 3 downscaling blocks (Figure 4). This creates a network shaped like U-Net, but functionally similar to DoubleCone. This aggressive downscaling reduces the number of operations necessary during inference, especially for larger maps (Table 7).

Instead of 1 value head, AnonymizedAI uses 3 values heads for 3 different value functions: (1) shaped reward similar to MicroRTS-Py except each combat unit type is scaled by build-time (rewarding expensive units more), (2) win-loss sparse reward at game end (Tanh activation), and (3) in-game difference in units based on cost (similar to the reward function used by Winter (2021)). These 3 value heads are used to mix-and-match rewards over the course of training, generally starting with dense rewards using heads (1) and (3) and finishing with only win-loss sparse rewards by the end.

### 3.2 BASE MODEL TRAINING

We train using the PPO loss function from Schulman et al. (2017):

$$L^{CLIP+VF+S}(\theta) = \hat{\mathbb{E}}_t \left[ L^{CLIP}(\theta) - c_1 L^{VF}(\theta) + c_2 S\left[\pi_\theta\right](s_t) \right], \tag{1}$$

$$L^{CLIP}(\theta) = \min\left( \frac{\pi_\theta(a_t|s_t)}{\pi_{\theta_{old}}(a_t|s_t)} \hat{A}_t, \text{clip}\left( \frac{\pi_\theta(a_t|s_t)}{\pi_{\theta_{old}}(a_t|s_t)}, 1-\epsilon, 1+\epsilon \right) \hat{A}_t \right), \tag{2}$$

$$L^{VF}(\theta) = \frac{1}{2}\left( V_\theta(s_t) - \hat{V}_t \right)^2, \tag{3}$$

where $c_1$ is the value loss coefficient, $c_2$ is the entropy coefficient, $S$ is an entropy bonus function, $\pi_\theta$ is the stochastic policy, $\pi_{\theta_{old}}$ is the rollout policy, $a_t$ is the action taken at time $t$, $s_t$ is the observation at time $t$, $\hat{A}_t$ is the advantage estimate, $\epsilon$ is the clipping coefficient, $V_\theta$ is the value function, $\hat{V}_t$ is the return estimate. We estimate the advantage using Generalized Advantage Estimation (GAE) (Schulman et al., 2016). We compute a separate advantage for each of the 3 rewards with independent $\gamma$ (discount factor) and $\lambda$ (exponential weight discount) for each reward. The 3 advantages are weighted summed together to get the final advantage estimate, which is used in computing the policy loss ($L^{CLIP}$).

The reward weights, value loss coefficients (each value head has its own loss coefficient), entropy coefficient, and learning rate are varied on a schedule (Table 8). At the start of training, the policy loss heavily weighs the shaped reward advantage, and the value loss similarly weighs the shaped value head. By the end of training, both losses are weighted more towards the win-loss sparse reward. The entropy coefficient is also lowered at the end of training to discourage the agent from making random moves as the learning rate is lowered. The schedule specifies phases where these values are set and transitions between phases by changing values linearly based on timesteps.

We first trained AnonymizedAI with self-play both against itself and against prior versions of itself (Table 9). We reached a 91% win rate against the same bots MicroRTS-Py was benchmarked against. However, it only beat CoacAI (the 2020 competition winner) in 20% of games. The best performing agent of MicroRTS-Py nearly always beat CoacAI; however, the best versions using GridNet also usually lost against CoacAI. We fine-tuned the model through 3 iterations: (1) one-half of environments trained against CoacAI (Table 10); (2) one-half of environments trained against CoacAI or Mayari (2021 competition winner) split evenly and primarily trained on win-loss rewards (Table 11); and (3) same as before with action mask improvements and a GELU activation after the stride-4 convolution to match Limburg (2023)'s DoubleCone. By the end of fine-tuning, the model was winning 98% of games, including about 90% against each of CoacAI and Mayari.

### 3.3 TRANSFER LEARNING

The model so far had been trained on the 5 smaller Open maps. Using the fine-tuned model as a starting point, we trained additional models, each trained exclusively on 1 of 3 Open maps. `NoWhereToRun9x8` is very different from the other maps with a wall of resources separating

opponents. `DoubleGame24x24` and `BWDistantResources32x32` are larger than the maps the base model trained on. All 3 transfer learning runs used the same schedule: (1) warmup of sparse, win-loss reward weights linearly transitioned to a mixture of both shaped and sparse rewards, (2) middle phase of mixed rewards, (3) end phase of sparse rewards again at a lower learning rate (Table 12). We did an additional fine-tuning on `NoWhereToRun9x8` using only the win-loss reward. These transfer learned agents exceeded 90% win-rate on their respective maps, significant improvements over the base model (especially on `BWDistantResources32x32` which started below 10%)

### 3.4 SQUNET TRAINING

We trained the squnet models with fewer steps because of time constraints and skipped the predominantly cost-based reward phase because it didn't help train the base model (Table 13). The 2 models (trained on maps of up to size 32x32 and 64x64) managed only a 40% win-rate, never beating CoacAI or Mayari. These models were policies of last resort.

We also fine-tuned the squnet-map32 model on only `BWDistantResources32x32` using the transfer learning sparse fine-tuning schedule. This fine-tuned model achieved 85% win-rate, beating Mayari half the time, but never beating CoacAI.

### 3.5 BEHAVIOR CLONING BOOTSTRAPPED TRAINING

In follow-up work after the AnonymizedAI submission, we wanted to train a model that (1) didn't require the shaped rewards and reward scheduling, (2) could be trained in fewer steps and less time, and (3) could defeat prior competition winners on the largest maps. We opted for a neural architecture between DoubleCone and squnet: a nested downscaling residual block each of stride 4, so the bottom block scales the input down to 1/16th the original size. At each downscaling level, there were multiple residual blocks (6 at full resolution, split evenly by the downscaling block; 4 at 1/4 resolution, split evenly by the 1/16 downscaling block; and 4 at 1/16 resolution) (Table 15). This architecture theoretically has a 128x128 receptive field while using 25% fewer operations than DoubleCone at inference time. On the largest Open map (`(4)BloodBath.scmB`), this is 6 times more computations than squnet-64. Therefore, this neural architecture won't be usable in a competition given the same hardware constraints as the 2023 competition.

Initially, we tried a similar training strategy to AnonymizedAI where the model is trained on 16x16 maps and that model is used for transfer learning to larger maps. However, we only managed a 60% win-rate on `BWDistantResources32x32` and less than a 20% win-rate on `(4)BloodBath.scmB` after over 100 million steps before terminating training.

Next, we tried imitation learning to bootstrap the model, similar to Vinyals et al. (2019). We got rid of the three rewards, opting for only the win-loss reward. microRTS doesn't have human replays, so we used playthroughs of the 2021 competition winner Mayari playing against itself, 2020 competition winner CoacAI, and POLightRush (baseline scripted bot and 2017 competition winner) (Table 16). Instead of generating an offline replay dataset, we set the microRTS environment to play bots against each other and these observations and actions were fed into rollouts used for behavior cloning the policy and fitting the value heads:

$$L^{BC+VF}(\theta) = \hat{\mathbb{E}}_t \left[ L^{BC}(\theta) - c_1 L^{VF}(\theta) \right], \tag{4}$$

$$L^{BC}(\theta) = -\frac{1}{\|a_t\|} \log \pi_\theta(a_t|s_t), \tag{5}$$

where $c_1$ and $L^{VF}$ are the same as in the PPO loss function. The behavior cloning policy loss is the cross-entropy loss between the policy logits and the actions taken by the Mayari bot. We found scaling the loss by the number of units accepting actions allowed the learning rate to be significantly increased. Scaling down the loss by the number of units keeps the losses for all turns roughly similar in scale as otherwise large unit count turns would have much larger losses as each unit's loss contribution is summed together (each unit's actions are assumed to be independent).

We trained 3 behavior cloned models (16x16, 32x32, and 64x64) on the same maps for each map size as AnonymizedAI training. The 64x64 model used the weights of the 32x32 model as a starting

point, while the other two models were randomly initialized. We then used PPO to fine-tune the behavior cloned models on the same maps (Table 17).

## 4  RESULTS

### 4.1  SINGLE PLAYER ROUND-ROBIN BENCHMARK

Table 1: Single player round-robin benchmark win rates. Win rates above 50% are bolded. (D) DoubleCone model used. (S) squnet model used.

| | WorkerRush | LightRush | CoacAI | Mayari | Overall |
|---|---|---|---|---|---|
| `basesWorkers8x8A` | **95** | **100** | **99** | **100** | **99** |
| `FourBasesWorkers8x8` | **100** | **100** | **100** | **98** | **100** |
| `NoWhereToRun9x8` | **100** | **100** | **93** | **99** | **98** |
| `basesWorkers16x16A` | **100** | **100** | **90** | **98** | **97** |
| `TwoBasesBarracks16x16` | **100** | **89** | **99** | **100** | **97** |
| `DoubleGame24x24` | **100** | **98** | **94** | **100** | **98** |
| `BWDistantResources32x32` | **99** (D) | **90** (D) | **88** (D) | **99** (D) | **94**[¶](D) |
| | **93** (S) | **73** (S) | 23 (S) | **58** (S) | **61** (S) |
| `(4)BloodBath.scmB` | **98** | 0 | 0 | 0 | 25[†] |
| **AI Average**[*] | **99** | **85** | **83** | **87** | **88** |

[*] AI Average uses the DoubleCone (D) results from `BWDistantResources32x32`.
[¶] AnonymizedAI lost 0.25% of matches (1 match) by timeout.
[†] AnonymizedAI lost 1% of matches (4 matches) by timeout.

In a single player round-robin benchmark on the Open maps (Table 1), AnonymizedAI beat the competition winners of 2021 (Mayari), 2020 (CoacAI), and 2017 (POLightRush, baseline) on 7 of the 8 maps (winning over 96% of games on these maps). AnonymizedAI could only beat the POWorkerRush baseline bot on the largest map, `(4)BloodBath.scmB`. The DistantResources fine-tuned squnet model performed worse than the DoubleCone model across all opponents, but maintained an over 50% win rate against all but CoacAI. Timeouts didn't affect results significantly.

### 4.2  IEEE-COG2023 MICRORTS COMPETITION RESULTS

The IEEE-CoG2023 microRTS competition is a round-robin tournament on 12 maps of different sizes and distributions of terrain, resources, and starting units and buildings. 8 Open maps are known beforehand, 4 Hidden maps are only revealed after the competition. The winner is the agent with the highest win rate on the 8 Open maps. Hidden map results are publicly available, but this paper will only discuss the Open maps. For this competition, a total of 11 agents were submitted: 9 programmatic policies, 1 synthesized programmatic policy, and AnonymizedAI. The competition also had 6 baselines: (1) RandomBiasedAI (performs actions randomly, biased towards attacking if able), (2) NaiveMCTS (a simple Monte Carlo tree search agent that searches until reaching the time limit) (3) POWorkerRush, (4) POLightRush, (5) 2L (programmatic strategies generated by Moraes et al. (2023), the competition organizers), and (6) the prior competition winner Mayari. The baselines cannot win the competition.

AnonymizedAI was declared the winner with the highest win rate (72%) across all submissions (Table 2). AnonymizedAI had a higher win rate than all but two baselines: 2L (76%) and Mayari (82%). AnonymizedAI had an over 50% win rate versus every opponent including 2L (60%) and Mayari (65%).

As expected from the single player round-robin benchmark, AnonymizedAI does better on smaller maps and dismally on the largest maps (Table 3). However, in the competition, AnonymizedAI underperformed against agents already benchmarked in the single player round-robin (14-19% lower win rate against each agent), even accounting for the likely use of the weaker squnet model on `BWDistantResources32x32`. Breaking down by map, AnonymizedAI underperformed against benchmarked agents by 20-40% on 5 maps.

Table 2: Win rates of selection of agents in the IEEE-CoG2023 microRTS competition. Player 1 is the row agent and player 2 is the column agent. Each win rate value is the percentage of games won by player 1. Cells are bolded if the win rate is higher than the opponent's row win rate. For example, AnonymizedAI vs ObiBotKenobi is bolded because 49% is higher than 47%, thus meaning the combined player 1 and 2 win rate is 51% for AnonymizedAI vs ObiBotKenobi. Overall includes all agents, including those not shown. Win rates for all agents are shown in Table 26.

| | Mayari | 2L | **AnonymizedAI** | ObiBotKenobi | POLightRush | POWorkerRush | Overall |
|---|---|---|---|---|---|---|---|
| Mayari (2021 winner) | - | **53** | 32 | **73** | **88** | **75** | 82 |
| 2L (baseline) | 51 | - | 39 | **50** | **75** | **88** | 76 |
| **AnonymizedAI** (2023 winner) | **62** | **59** | - | 49 | **64** | **78** | 72 |
| ObiBotKenobi (2023 2nd place) | 39 | 29 | 47 | - | **58** | **65** | 66 |
| POLightRush (baseline) | 0 | 25 | 29 | 38 | - | **69** | 55 |
| POWorkerRush (baseline) | 13 | 13 | 21 | 29 | 38 | - | 53 |

Table 3: AnonymizedAI win rates in 2023 competition by opponent and map. Bolded cells are win rates over 50%. Overall includes all agents, including those not shown.

| | POWorkerRush | POLightRush | ObiBotKenobi | 2L | Mayari | Overall |
|---|---|---|---|---|---|---|
| basesWorkers8x8A | **60** | **70** | **60** | **60** | **60** | **66** |
| FourBasesWorkers8x8 | **100** | **100** | 20 | **95** | **100** | **95** |
| NoWhereToRun9x8 | **90** | **85** | **83** | **70** | **70** | **84** |
| basesWorkers16x16A | **100** | **100** | **95** | **100** | **100** | **100** |
| TwoBasesBarracks16x16 | **80** | **80** | 10 | **70** | **80** | **75** |
| DoubleGame24x24 | **80** | **75** | **78** | **80** | **75** | **80** |
| BWDistantResources32x32 | 50 | 30 | 35 | 3 | 35 | **54** |
| (4)BloodBath.scmB | **70** | 0 | 28 | 0 | 0 | 34 |
| AI Average | **79** | **68** | **51** | **60** | **65** | **74** |

The competition ran jobs, splitting each map into 5 or 10 jobs where each job would run a complete round-robin with all agents on that map playing 2 or 1 games as player 1 and 2 each. For basesWorkers8x8A, on which AnonymizedAI underperformed by almost 40%, the competition had 5 jobs. On the first 3 jobs, AnonymizedAI won nearly every game. On the last 2 jobs, AnonymizedAI lost nearly every game. 1 or 2 jobs per underperforming map appear to have outlier low win rates for AnonymizedAI (Table 27).

### 4.3 BEHAVIOR CLONING RESULTS

We created 2 additional agents from the behavior cloning (BC-Agent) and the following PPO fine-tuning (BC-PPO-Agent). Each agent consisted of the models trained on their respective map sizes (16x16, 32x32, and 64x64). These agents do not have any map-specific models.

BC-Agent had a 71% win rate, doing well against the POLightRush baseline (96%, better than AnonymizedAI) and respectably against Mayari (44%) (Table 28). On the largest map, BC-Agent manages to occasionally beat POLightRush (63%), CoacAI (20%), and Mayari (40%) compared to AnonymizedAI's 0% across all 3 opponents.

Table 4: BC-PPO-Agent win rate in a single player round-robin benchmark. Win rates above 50% are bolded.

|  | POWorkerRush | POLightRush | CoacAI | Mayari | Overall |
|---|---|---|---|---|---|
| `basesWorkers8x8A` | **92** | **100** | **85** | **100** | **94** |
| `FourBasesWorkers8x8` | **100** | **100** | **100** | **100** | **100** |
| `NoWhereToRun9x8` | **100** | **100** | **90** | **80** | **92** |
| `basesWorkers16x16A` | **100** | **100** | **95** | **95** | **98** |
| `TwoBasesBarracks16x16` | **100** | 0 | **100** | **95** | **74** |
| `DoubleGame24x24` | **98** | **85** | **100** | **100** | **96** |
| `BWDistantResources32x32` | **100** | **100** | **95** | **100** | **99** |
| `(4)BloodBath.scmB` | **100** | **88** | 0 | 5 | 48 |
| AI Average | **99** | **84** | **83** | **84** | **88** |

Once fine-tuned with PPO, BC-PPO-Agent obtains an AnonymizedAI-comparable 88% win rate (Table 4). BC-PPO-Agent generally improves upon BC-Agent's win rates on each map and against each opponent. However, the biggest exceptions are POLightRush on `TwoBasesBarracks16x16` (from 100% to 0%) and the largest map where the fine-tuned model can no longer beat CoacAI and Mayari.

While AnonymizedAI required map-specific fine-tuned models to be competitive on `NoWhereToRun9x8`, `DoubleGame24x24`, and `BWDistantResources32x32`, BC-PPO-Agent only has models for the different map-sizes. This demonstrates promise for creating generalized agents that can play across a wide range of maps.

## 5 DISCUSSION

### 5.1 IMPROVING INFERENCE TIME IN MICRORTS COMPETITIONS

AnonymizedAI's underperformance in the 2023 competition suggests that job environments can run slow. We worked with the competition organizers to reduce the chance of timeouts. However, it was difficult to reproduce the same results as the competition servers in our development environments.

Improving inference time is critical to matching benchmark results in a competition. We suggest 3 improvements (2 for agents, and 1 for the competition organizers): (1) use fast inference runtime providers like OpenVINO for ONNX Runtime, (2) continue to train agents using the smaller squnet models (possibly with behavior cloning to bootstrap training), and (3) replace the fixed per turn timeout tolerance in the competition with an overtime budget. For DoubleCone, Limburg (2023) found using OpenVINO could have made inference 2-3 times faster in the LUX competition. This would likely make running DoubleCone or the larger squnet models on all maps except the largest feasible for the competition. An overtime budget for an entire match instead of the 20 ms per turn tolerance will help agents deal with environment instabilities. For example, BC-PPO-Agent timed out in 11% of games on `(4)BloodBath.scmB`, despite averaging 55 ms/turn and going over 100 ms in only 0.016% of turns (averaging less than 1 over 100 ms turn per game). An overage budget of even 1 second per game would likely prevent most timeouts.

### 5.2 TRAINING ON LARGER MAPS

None of our agents managed to reliably defeat the prior two competition winners on the largest map. `(4)BloodBath.scmB` is a challenging map for DRL because game lengths are significantly longer than on smaller maps. BC-PPO-Agent averaged 3,500 steps per game on `(4)BloodBath.scmB` compared to around 925 steps on the next largest map, `BWDistantResources32x32`. DRL must learn to propagate rewards over longer time periods, and the observation-action space is so large that DRL can only hope to explore a fraction of it. A rushing strategy of sending attack units and surplus workers towards the enemy as soon as

possible is a strong strategy on all but the largest map. (4)BloodBath.scmB seems to reward a build up of forces before attacking, which DRL struggled to learn.

We hoped imitation learning would mitigate these issues by providing a model that generates non-zero win-loss rewards and reasonable observation-action pairs. However, during PPO fine-tuning, a training policy that initially won 40-50% of training games, dropped to 20% midway through training. It eventually recovers to winning 40% of training games; however, the fine-tuned policy had a worse evaluation win rate than the initial supervised policy. This training curve differs from the smaller map fine-tuning where the training policy quickly won 60% of training games and improved upon the evaluation win rate.

PPO (and DRL algorithms in general) have many hyperparameters that need to be tuned before a model is able to learn, and our hyperparameter search is ongoing. Other algorithms might perform better from a behavior cloned start. For example, Advantage Actor Critic (A2C) (Mnih et al., 2016) uses a loss function similar to behavior cloning, and thus policy training might more smoothly transition from behavior cloning to DRL.

Extending the curriculum to include more game states could also improve large map training. For example, varied existing agents can be used to advance the game before switching to a training agent to finish the game (Uchendu et al., 2023). Another way is to utilize a prioritized fictitious self-play mechanism used by Vinyals et al. (2019) that prioritizes training the agent on the most difficult prior agent checkpoints.

### 5.3 Behavior cloning, transfer learning, and academic competitions

Training multiple models for AnonymizedAI took 70 GPU-days. Imitation learning BC-Agent trained for 23 GPU-days, and PPO fine-tuning BC-PPO-Agent took another 49 GPU-days. These are significant amounts of compute for a mostly academic competition.

There are several ways to make DRL more feasible in a competition and educational setting: (1) focus on smaller maps, (2) fine-tune pretrained models from DRL or behavior cloning, (3) transfer an existing model to new maps, or (4) use a significantly smaller neural architecture. The largest map took 19 GPU-days to train for AnonymizedAI, 15 GPU-days for BC-Agent, and 34 GPU-days for BC-PPO-Agent. Over two-thirds of training time for BC-PPO-Agent was spent training on the largest map to little benefit. Huang et al. (2021) trained an agent for player 1 on a single 16x16 map in 60 hours. BC-PPO-Agent trained on the 5 Open maps up to size 16x16 in about 7.5 days.

Fine-tuning and transfer learning were critical to making AnonymizedAI competitive. Both took significantly less time than training from randomly initialized weights because the policy already makes reasonable tactical actions and the critic already makes reasonable value estimates. If future competitions change the Open maps, fine-tuning and transfer learning will significantly help AnonymizedAI and BC-PPO-Agent. We didn't train BC-PPO-Agent on specific maps, so fine-tuning there could improve win rates. Behavior cloning other agents (possibly several agents simultaneously), will bootstrap DRL agents to effective policies that would be extremely difficult to obtain with naive DRL training.

AnonymizedAI's DoubleCone and BC-PPO-Agent's deep squnet are relatively large neural networks, each at around 5 million parameters. Huang et al. (2021)'s best performing policies each used fewer than 1 million parameters. These smaller networks are quicker to train and have faster inference time, and there's no definitive evidence so far that they are worse than larger networks.

## 6 Conclusion

AnonymizedAI is the first DRL agent to win a microRTS competition. It demonstrates that an iterative training process of fine-tuning and transfer learning is effective for creating competitive DRL agents. Such a training process can be used by resource-constrained researchers and students to create novel DRL agents for future competitions and experiments. Fine-tuning behavior cloning with PPO is a promising way to create competitive DRL agents without needing to handcraft shaped reward functions.

## 7 REPRODUCIBILITY STATEMENT

Our Methods section describes the architectures and training processes for AnonymizedAI, BC-Agent, and BC-PPO-Agent. We give additional details in Appendices: Appendix B neural network architectures, Appendix C initial training details, Appendix D transfer learning details, Appendix E squnet learning details, and Appendix F behavior cloning details. Appendix H describes the setup of the single player round-robin tournaments for AnonymizedAI (Section 4.1), BC-Agent, and BC-PPO-Agent (Section 4.3). The microRTS competition requires agents to be open sourced. Our open-sourced code repository includes a link to the archive used for the competition and instructions on how to run the submission in the competition environment[5]. Supplementary Material includes an anonymized code repository; however, links had to be removed or broken.

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

## A  COMPETITION DETAILS

Table 5: Policy networks used by AnonymizedAI

| Network | Usage |
| --- | --- |
| ppo-Microrts-finetuned-NoWhereToRun-S1-best | NoWhereToRun9x8 |
| ppo-Microrts-A6000-finetuned-coac-mayari-S1-best | All other maps of size 16x16 and smaller |
| ppo-Microrts-finetuned-DoubleGame-shaped-S1-best | DoubleGame24x24 |
| ppo-Microrts-finetuned-DistantResources-shaped-S1-best | BWDistantResources32x32 if completion time under 75 ms |
| ppo-Microrts-squnet-DistantResources-128ch-finetuned-S1-best | BWDistantResources32x32 if completion time above 75 ms |
| ppo-Microrts-squnet-map32-128ch-selfplay-S1-best | All other maps where longest dimension is between 17-32 |
| ppo-Microrts-squnet-map64-64ch-selfplay-S1-best | Maps where the longest dimension is over 32 |

To participate in the competition, AnonymizedAI has a Java class that handles turn handling and resetting commands from the Java game engine. While earlier Python solutions passed JSON or XML data over a socket[6], AnonymizedAI passes binary data over a pipe to the Python process as a performance optimization for the larger maps.

Each agent played every other agent on each map 20 times (10 each as player 1 and 2). Timeouts were disabled for the competition, but the Java-side of AnonymizedAI would skip its turn (submitting no actions) if 100 ms had elapsed. On `BWDistantResources32x32`, AnonymizedAI chose between the DoubleCone and squnet fine-tuned models by running both models on the first observation 100 times each and choosing DoubleCone if it computed actions within 75 ms on average.

---

[6]https://github.com/douglasrizzo/python-microRTS

Table 6: Open competition maps. Representation column is the size of the vectorized observation in AnonymizedAI.

| Name | Size | Representation | Start |
|------|------|----------------|-------|
| basesWorkers8x8A | 8x8 | 16x16 |  |
| FourBasesWorkers8x8 | 8x8 | 16x16 |  |
| NoWhereToRun9x8 | 9x8 | 12x12*or 16x16¶ |  |
| basesWorkers16x16A | 16x16 | 16x16 |  |
| TwoBasesBarracks16x16 | 16x16 | 16x16 |  |
| DoubleGame24x24 | 24x24 | 24x24†or 32x32‡ |  |
| BWDistantResources32x32 | 32x32 | 32x32 |  |
| (4)BloodBath.scmB | 64x64 | 64x64 |  |

* ppo-Microrts-finetuned-NoWhereToRun-S1-best uses a 12x12 representation
¶ ppo-Microrts-A6000-finetuned-coac-mayari-S1-best
† ppo-Microrts-finetuned-DoubleGame-shaped-S1-best uses a 24x24 representation.
‡ ppo-Microrts-squnet-map32-128ch-selfplay-S1-best pads the observation to 32x32.

## B  NEURAL NETWORK ARCHITECTURE

DoubleCone(4, 6, 4) ((Limburg, 2023)) consists of (1) 4 residual blocks; (2) a downscaled residual block consisting of a stride-4 convolution, 6 residual blocks, and 2 stride-2 transpose convolutions; (3) 4 residual blocks; and (4) actor and value heads (Figure 1). Each residual block includes a squeeze-excitation layer after the second convolutional layer (Figure 2). The values heads are each (1) 2 stride-2 convolutions, (2) an adaptive average pooling layer, (3) flattened, (4) 2 densely connected layers, and (5) an activation function (Identity [no activation] or Tanh) to a single, scalar value (Figure 3). The adaptive average pooling layer allows the network to be used on different map sizes.

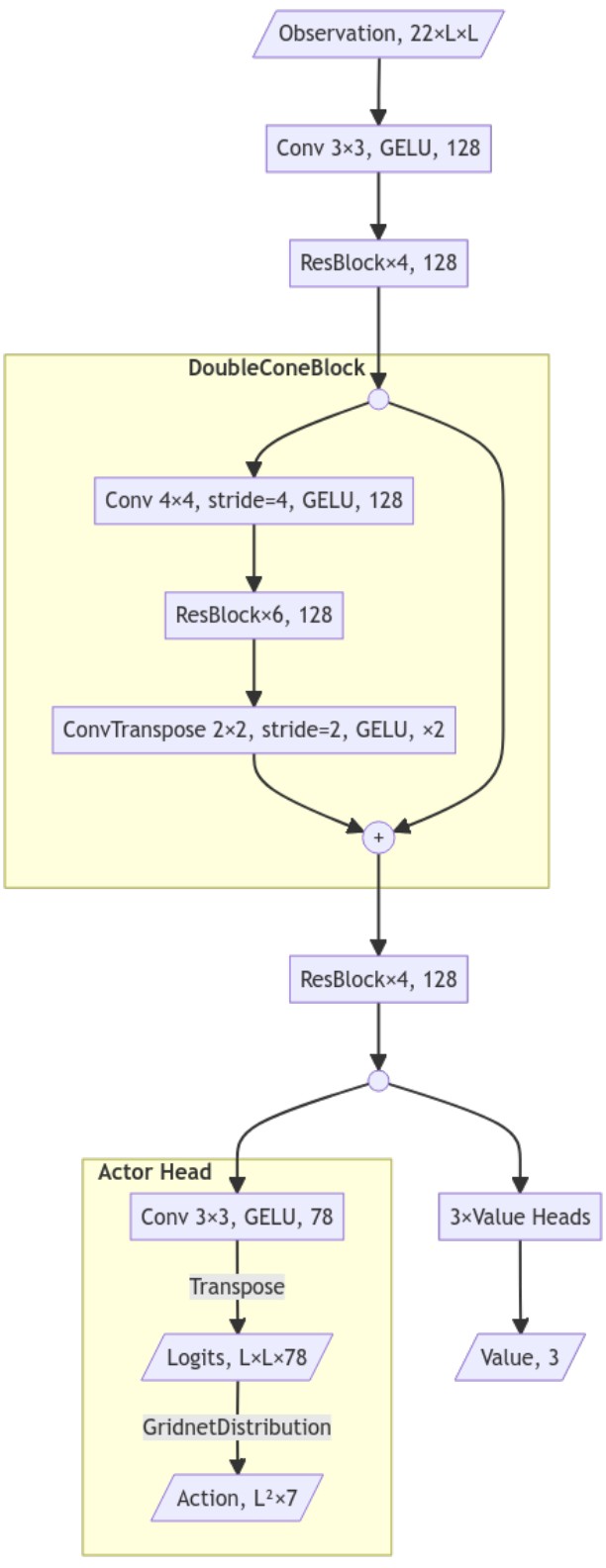

Figure 1: DoubleCone(4, 6, 4) neural network architecture.

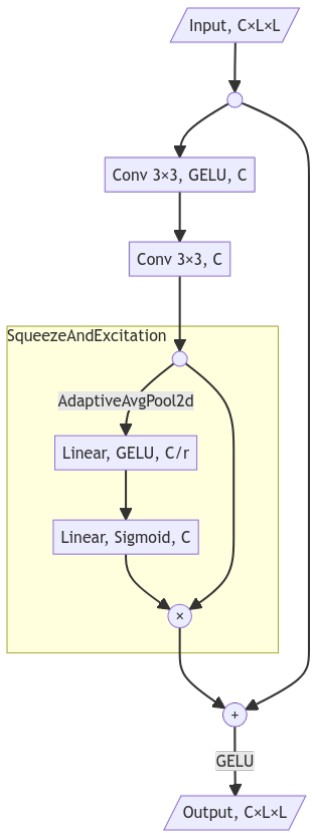

Figure 2: ResBlock used in DoubleCone, squnet32, and squnet64. The residual block is similar to a standard residual block but inserts a Squeeze-Excitation block after the convolutional layers and before the residual connection.

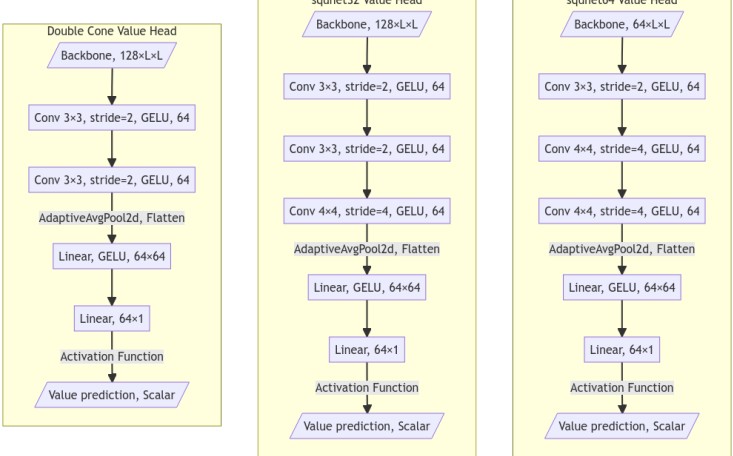

Figure 3: Value heads used in (from left to right) DoubleCone, squnet32, and squnet64. The AdaptiveAvgPool2d layer allows the network to be used on other map sizes.

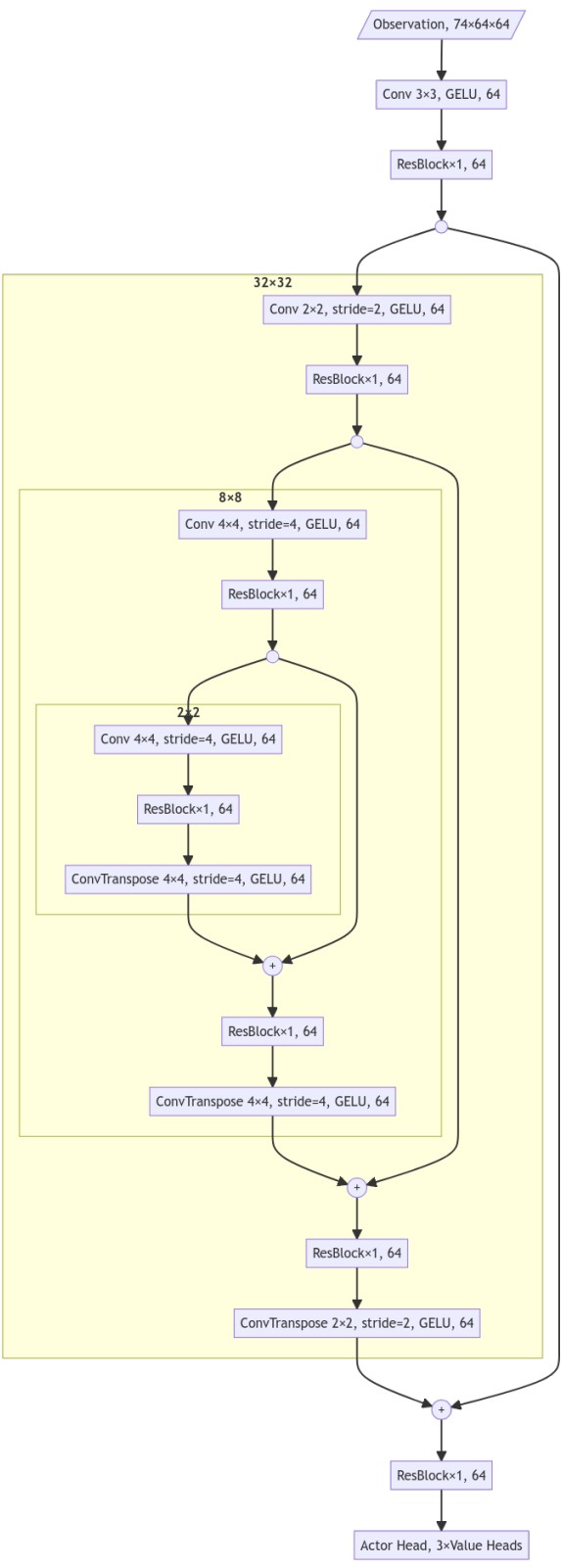

Figure 4: squnet64 neural network architecture. Instead of one downscaling block as in DoubleCone, this network downscales 3 times. This aggressive downscaling reduces the number of computations for larger maps, while theoretically supporting a large receptive field.

Table 7: Comparison of different architectures

|  | **DoubleCone** | **squnet-map32[¶]** | **squnet-map64** |
|---|---|---|---|
| Levels | 2 | 4 | 4 |
| Encoder residual blocks/level | [4, 6] | [1, 1, 1, 1] | [1, 1, 1, 1] |
| Decoder residual blocks/level | [4] | [1, 1, 1] | [1, 1, 1] |
| Stride/level | [4] | [2, 2, 4] | [2, 4, 4] |
| Deconvolution strides/level | [[2, 2]][*] | [2, 2, 4] | [2, 4, 4] |
| Channels/level | [128, 128] | [128, 128, 128, 128] | [64, 64, 64, 64] |
| Trainable parameters | 5,014,865 | 3,584,657 | 1,420,625 |
| MACs[†] | 0.70B (16x16)[‡]
0.40B (12x12)[§]
1.58B (24x24)
2.81B (32x32) | 1.16B (32x32) | 1.41B (64x64) |

[¶] Used by ppo-Microrts-squnet-DistantResources-128ch-finetuned-S1-best and ppo-Microrts-squnet-map32-128ch-selfplay-S1-best.

[*] 2 stride-2 transpose convolutions to match the 1 stride-4 convolution.

[†] Multiply-Accumulates for computing actions for a single observation.

[‡] All maps smaller than 16x16 (except NoWhereToRun9x8) are padded with walls up to 16x16.

[§] NoWhereToRun9x8 is padded with walls up to 12x12.

## C    INITIAL TRAINING DETAILS

AnonymizedAI was trained with partial observability and environment non-determinism disabled.

Table 8: Initial training schedule from a randomly initialized model

|  | Phase 1 | Transition 1→2[*] | Phase 2 | Transition 2→3[*] | Phase 3 |
|---|---|---|---|---|---|
| steps | 90M | 60M | 30M | 60M | 60M |
| reward weights[†] | [0.8, 0.01, 0.19] |  | [0, 0.5, 0.5] |  | [0, 0.99, 0.01] |
| $c_1$ (value loss coef)[†] | [0.5, 0.1, 0.2] |  | [0, 0.4, 0.4] |  | [0, 0.5, 0.1] |
| $c_2$ (entropy coef) | 0.01 |  | 0.01 |  | 0.001 |
| learning rate | $10^{-4}$ |  | $10^{-4}$ |  | $5 \times 10^{-5}$ |

[*] Values are linearly interpolated between phases based on step count.

[†] Listed weights are for the shaped, win-loss, cost-based values, respectively.

Table 9: Comparison of initial training, shaped fine-tuning, and sparse fine-tuning parameters

| Parameter | Initial Training | Shaped Fine-Tuning | Sparse Fine-Tuning |
|---|---|---|---|
| Steps | 300M | 100M | 100M |
| Number of Environments | 24 | " | " |
| Rollout Steps Per Env | 512 | " | " |
| Minibatch Size | 4096 | " | " |
| Epochs Per Rollout | 2 | " | " |
| $\gamma$ (Discount Factor) | [0.99, 0.999, 0.999][*] | " | " |
| GAE $\lambda$ | [0.95, 0.99, 0.99][¶] | " | " |
| Clip Range | 0.1 | " | " |
| Clip Range VF | 0.1 | " | " |
| VF Coef Halving[‡] | True | " | " |
| Max Grad Norm | 0.5 | " | " |
| Latest Self-play Envs | 12 | " | " |
| Old Self-play Envs | 12 | 0 | 0 |
| Bots | none | CoacAI: 12 | CoacAI: 6 
 Mayari: 6 |
| Maps | basesWorkers16x16A 
 TwoBasesBarracks16x16 
 basesWorkers8x8A 
 FourBasesWorkers8x8 
 NoWhereToRun9x8 
 EightBasesWorkers16x16[†] | " | " |

["] Same value as cell to left.
[*] Value per value head (shaped, win-loss, cost-based).
[¶] Multiply v_loss by 0.5, as done in CleanRL.
[†] Map not used in competition.

Table 10: Shaped fine-tuning schedule

| | Start | Transition →1[*] | Phase 1 | Transition 1→2[*] | Phase 2 |
|---|---|---|---|---|---|
| steps | | 5M | 30M | 20M | 45M |
| reward weights[†] | [0, 0.99, 0.01] | | [0, 0.5, 0.5] | | [0, 0.99, 0.01] |
| $c_1$ (value loss coef)[†] | [0, 0.4, 0.2] | | [0, 0.4, 0.4] | | [0, 0.5, 0.1] |
| $c_2$ (entropy coef) | 0.01 | | 0.01 | | 0.001 |
| learning rate | $10^{-5}$ | | $5 \times 10^{-5}$ | | $5 \times 10^{-5}$ |

[*] Values are linearly interpolated between phases based on step count.
[†] Listed weights are for the shaped, win-loss, cost-based values, respectively.

Table 11: Sparse fine-tuning schedule

| | Phase 1 | Transition 1→2[*] | Phase 2 |
|---|---|---|---|
| steps | 30M | 40M | 30M |
| reward weights[†] | [0, 0.99, 0.01] | | [0, 0.99, 0.01] |
| $c_1$ (value loss coef)[†] | [0, 0.5, 0.1] | | [0, 0.5, 0.1] |
| $c_2$ (entropy coef)) | 0.001 | | 0.0001 |
| learning rate | $5 \times 10^{-5}$ | | $10^{-5}$ |

[*] Values are linearly interpolated between phases based on step count.
[†] Listed weights are for the shaped, win-loss, cost-based values, respectively.

# D    TRANSFER LEARNING DETAILS

Table 12: Transfer learning schedule starting from ppo-Microrts-A6000-finetuned-coac-mayari-S1-best model

|  | Start | Transition →1[*] | Phase 1 | Transition 1→2[*] | Phase 2 |
|---|---|---|---|---|---|
| steps |  | 5M | 30M | 20M | 45M |
| reward weights[†] | [0, 0.99, 0.01] |  | [0.4, 0.5, 0.1] |  | [0, 0.99, 0.01] |
| $c_1$ (value loss coef)[†] | [0.2, 0.4, 0.2] |  | [0.3, 0.4, 0.1] |  | [0, 0.5, 0.1] |
| $c_2$ (entropy coef) | 0.01 |  | 0.01 |  | 0.0001 |
| learning rate | $5 \times 10^{-5}$ |  | $7 \times 10^{-5}$ |  | $10^{-5}$ |

[*] Values are linearly interpolated between phases based on step count.
[†] Listed weights are for the shaped, win-loss, cost-based values, respectively.

# E    SQUNET LEARNING DETAILS

Table 13: Squnet training parameters

| Parameter | map32 | map32-DistantResources | map64 |
|---|---|---|---|
| Steps | 200M | 100M | 200M |
| n_envs | 24 | " | " |
| Rollout Steps Per Env | 512 | 512 | 256 |
| Minibatch Size | 2048 | 2048 | 258 |
| Clip Range | 0.1 | " | " |
| Clip Range VF | none | " | " |
| Latest Self-play Envs | 12 | " | " |
| Old Self-play Envs | 6 | 6 | 4 |
| Bots | CoacAI: 3 
 Mayari: 3 | CoacAI: 3 
 Mayari: 3 | CoacAI: 4 
 Mayari: 4 |
| Maps | DoubleGame24x24 
 BWDistantResources32x32 
 chambers32x32[*] | BWDistantResources32x32 | BloodBath.scmB 
 BloodBath.scmE[*] |

["] Same value as cell to left.
[*] Not competition Open maps.

Table 14: squnet training schedule starting with randomly initialized weights

|  | Phase 1 | Transition 1→2[*] | Phase 2 |
|---|---|---|---|
| steps | 100M | 60M | 40M |
| reward weights[†] | [0.8, 0.01, 0.19] |  | [0, 0.99, 0.01] |
| $c_1$ (value loss coef)[†] | [0.5, 0.1, 0.2] |  | [0, 0.5, 0.1] |
| $c_2$ (entropy coef) | 0.01 |  | 0.001 |
| learning rate | $10^{-4}$ |  | $5 \times 10^{-5}$ |

[*] Values are linearly interpolated between phases based on step count.
[†] Listed weights are for the shaped, win-loss, cost-based values, respectively.

# F  BEHAVIOR CLONING DETAILS

Table 15: Neural architecture for behavior cloning and PPO fine-tuned training

|  | deep16-128 |
|---|---|
| Levels | 3 |
| Encoder residual blocks/level | [3, 2, 4] |
| Decoder residual blocks/level | [3, 2] |
| Stride per level | [4, 4] |
| Deconvolution strides per level | [[2, 2], [2, 2]]* |
| Channels per level | [128, 128, 128] |
| Trainable parameters | 5,027,279 |
| MACs[†](16x16) | 0.52B |
| MACs[†](64x64) | 8.40B |

\* 2 stride-2 transpose convolutions to match the 1 stride-4 convolution.
[†] Multiply-Accumulates for computing actions for a single observation.

Table 16: Behavior cloning training parameters. " means same value as the cell to the left.

| Map Size | 16x16 | 32x32 | 64x64 |
|---|---|---|---|
| Steps | 100M | " | " |
| Number of Environments | 36 | 24 | 24 |
| Rollout Steps Per Env | 512 | " | " |
| Minibatch Size | 3072 | 768 | 192 |
| Epochs Per Rollout | 2 | " | " |
| $\gamma$ (Discount Factor) | 0.999 | 0.9996 | 0.999 |
| GAE $\lambda$ | 0.99 | 0.996 | 0.999 |
| Max Grad Norm | 0.5 | " | " |
| Gradient Accumulation | FALSE | FALSE | TRUE |
| Scale Loss by # Actions | TRUE | " | " |
| Bots | Mayari: 12
CoacAI: 12
POLightRush: 12 | Mayari: 12
CoacAI: 6
POLightRush: 6 | Mayari: 8
CoacAI: 8
POLightRush: 8 |
| Maps | basesWorkers16x16A
TwoBasesBarracks16x16
basesWorkers8x8A
FourBasesWorkers8x8
NoWhereToRun9x8
EightBasesWorkers16x16 | DoubleGame24x24
BWDistantResources32x32
chambers32x32 | (4)BloodBath.scmB
(4)BloodBath.scmE |

Table 17: Training parameters for PPO of behavior cloned models. " means same value as the cell to the left.

| Map Size | 16x16 | 32x32 | 64x64 |
|---|---|---|---|
| Steps | 100M | 200M | 200M |
| Number of Environments | 36 | 24 | 48 |
| Rollout Steps Per Env | 512 | " | " |
| Minibatch Size | 3072 | 768 | 192 |
| Epochs Per Rollout | 2 | " | " |
| $\gamma$ (Discount Factor) | 0.999 | 0.9996 | 0.99983 |
| GAE $\lambda$ | 0.99 | 0.996 | 0.9983 |
| Clip Range | 0.1 | " | " |
| Clip Range VF | none | " | " |
| VF Coef Halving‡ | TRUE | " | " |
| Max Grad Norm | 0.5 | " | " |
| Gradient Accumulation | FALSE | TRUE | TRUE |
| Latest Selfplay Envs | 12 | 12 | 28 |
| Old Selfplay Envs | 12 | 6 | 12 |
| Bots | Mayari: 6 CoacAI: 6 | Mayari: 3 CoacAI: 3 | Mayari: 2 CoacAI: 2 POLightRush: 2 POWorkerRush: 2 |
| Maps | basesWorkers16x16A TwoBasesBarracks16x16 basesWorkers8x8A FourBasesWorkers8x8 NoWhereToRun9x8 EightBasesWorkers16x16 | DoubleGame24x24 BWDistantResources32x32 chambers32x32 | (4)BloodBath.scmB (4)BloodBath.scmE |

Table 18: Behavior cloning schedule for 16x16 maps. Values in transition are linearly interpolated.

| | Start | Transition | End |
|---|---|---|---|
| | | 100M | |
| learning rate | $8 \times 10^{-5}$ | | 0 |

Table 19: Behavior cloning schedule for 32x32 and 64x64 maps. Values in transitions are cosine interpolated.

| | Start | Transition $\rightarrow$1 | Phase 1 | Transition 1$\rightarrow$2 | Phase 2 |
|---|---|---|---|---|---|
| | | 5M | 5M | 85M | 5M |
| learning rate | $10^{-5}$ | | $8 \times 10^{-5}$ | | $10^{-6}$ |

Table 20: Schedule for PPO fine-tuning of behavior cloned model for 16x16 map. Transition values are cosine interpolated.

| | Start | Transition $\rightarrow$1 | Phase 1 | Transition 1$\rightarrow$2 | Phase 2 |
|---|---|---|---|---|---|
| | | 5M | 5M | 85M | 5M |
| $c_2$ (entropy coef) | 0.001 | | 0.001 | | 0.0001 |
| learning rate | $10^{-5}$ | | $5 \times 10^{-5}$ | | $10^{-5}$ |

Table 21: Schedule for PPO fine-tuning of behavior cloned model for 32x32 map. Transition values are cosine interpolated.

|  | Start | Transition →1 | Phase 1 | Transition 1→2 | Phase 2 |
|---|---|---|---|---|---|
|  |  | 10M | 80M | 70M | 40M |
| $c_2$ (entropy coef) | 0.001 |  | 0.001 |  | 0.0001 |
| learning rate | $10^{-5}$ |  | $5 \times 10^{-5}$ |  | $10^{-5}$ |

Table 22: Schedule for PPO fine-tuning of behavior cloned model for 64x64 map. Transition values are cosine interpolated. Transition 1→2 being empty means values jump from Phase 1 to Phase 2.

|  | Start | Transition →1 | Phase 1 | Transition 1→2 | Phase 2 | Transition 2→3 | Phase 3 | Transition 3→4 | Phase 4 |
|---|---|---|---|---|---|---|---|---|---|
|  |  | 10M |  |  |  | 40M | 80M | 66M | 4M |
| $c_2$ (entropy coef) | 0 |  | 0 |  | 0.001 |  | 0.001 |  | 0.0001 |
| learning rate | $10^{-6}$ |  | $5 \times 10^{-5}$ |  | $10^{-6}$ |  | $5 \times 10^{-5}$ |  | $10^{-6}$ |
| freeze backbone and policy head | TRUE |  | TRUE |  | FALSE |  | FALSE |  | FALSE |

# G    TRAINING DURATIONS

We trained using Lambda Labs GPU on-demand instances. We used single Nvidia GPU instances, but different ones to be able to fit larger minibatches onto the GPU. A10 (24 GB VRAM) and A100 (40 GB VRAM) machines had 30 vCPUs and 200 GB RAM. A6000 (48 GB VRAM) machines had 14 vCPUs and 46 GB RAM. We did not fully utilize the CPU, RAM, or hard drive resources during training.

Behavior cloning and PPO fine-tuning of behavior cloned models were trained only using A10 machines. We had implemented gradient accumulation at this point to support larger batch sizes that did not need to fit on the GPU all-at-once.

Table 23: AnonymizedAI training durations. Blank models are intermediate models that lead to the next row. For example, the first 3 runs are intermediate models for 16x16. Runs are uploaded to the Anonymized benchmark Weights and Biases project, except for squnet-DistantResources (Anonymized microRTS Weights and Biases project path).

| Map | Run ID | GPU | Days Training |
|---|---|---|---|
| | df4flrs4 | A10 | 12.5 |
| | 9bz7wsuv | A6000 | 2.7 |
| | tff7xk4b | A6000 | 4.1 |
| 16x16 | 1ilo9yae | A6000 | 4.3 |
| | hpp5pffx | A10 | 1.9 |
| NoWhereToRun9x8 | vmns9sbe | A10 | 1.7 |
| DoubleGame24x24 | unnxtprk | A6000 | 5.3 |
| BWDistantResources32x32 | x4tg80vk | A100 | 3.6 |
| 32x32 | tga53t25 | A6000 | 10.2 |
| squnet-DistantResources | jl8zkpfr | A6000 | 5.0 |
| 64x64 | nh5pdv4o | A6000 | 19.0 |
| | | | 70.4 |

Table 24: Behavior cloning training durations. Runs are uploaded to the Anonymized microRTS Weights and Biases project.

| Map Size | Run ID | Days Training |
|---|---|---|
| 16x16 | lhs1b2gj | 3.5 |
| 32x32 | 16o4391r | 4.7 |
| 64x64 | uksp6znl | 15.1 |
| | | 23.3 |

Table 25: Training durations for PPO fine-tuning of behavior cloned models. Runs are uploaded to the Anonymized microRTS Weights and Biases project.

| Map Size | Run ID | Days Training |
|---|---|---|
| 16x16 | a4efzeug | 4.0 |
| 32x32 | 042rwd8p | 11.3 |
| 64x64 | 9l2debnz | 33.9 |
| | | 49.1 |

## H SINGLE PLAYER ROUND-ROBIN BENCHMARK SETUP

In Section 4.1, AnonymizedAI plays on the 8 Open maps against 4 opponents: (1) baseline POWorkerRush, (2) baseline and 2017 competition winner POLightRush, (3) 2020 competition winner CoacAI, and (4) last competition (2021) winner Mayari. AnonymizedAI normally plays against each opponent on each map for 100 matches (50 each as player 1 and 2). The exception is squnet (S) on BWDistantResources32x32, where AnonymizedAI only played each opponent for 20 matches (10 each as player 1 and 2). All opponents use A* for pathfinding, which is default for competitions. Win rates are percentages of wins where draws count as 0.5 wins for each player. The single player round-robin benchmark was run on a 2018 Mac Mini with Intel i7-8700B CPU (6-core, 3.2GHz) with PyTorch limited to 6 threads. Timeouts were set to 100 ms. If an agent took 20ms over the deadline (120 ms total), the game was terminated and the win awarded to the opponent.

In Section 4.3, BC-Agent and BC-PPO-Agent play each opponent on each map in 20 only games (10 each as player 1 and 2).

# I  ADDITIONAL IEEE-CoG2023 MICRORTS COMPETITION DETAILS

Table 26: Win rates of all agents in the IEEE-CoG 2023 microRTS competition on Open maps. Player 1 is the row agent and player 2 is the column agent. Each win rate value is the percentage of games won by player 1 (the row agent).

| | Mayari | 2L | AnonymizedAI | ObiBotKenobi | Aggrobot | sophia | bRHEAdBot | Ragnar | POLightRush | SaveTheBeesV4 | POWorkerRush | MyMicroRtsBot | NaiveMCTS | myBot | NIlSiBot | Predator | RandomBiasedAI | Overall |
|---|---|---|---|---|---|---|---|---|---|---|---|---|---|---|---|---|---|---|
| Mayari | - | 53 | 32 | 73 | 78 | 93 | 95 | 64 | 88 | 93 | 75 | 78 | 100 | 100 | 100 | 100 | 100 | 82 |
| 2L | 51 | - | 39 | 50 | 69 | 63 | 93 | 56 | 75 | 98 | 88 | 81 | 76 | 94 | 94 | 95 | 96 | 76 |
| **AnonymizedAI** | 62 | 59 | - | 49 | 64 | 71 | 64 | 64 | 64 | 78 | 78 | 76 | 84 | 94 | 73 | 87 | 87 | 72 |
| ObiBotKenobi | 39 | 29 | 47 | - | 47 | 69 | 60 | 56 | 58 | 83 | 65 | 76 | 72 | 99 | 79 | 85 | 100 | 66 |
| Aggrobot | 9 | 25 | 26 | 60 | - | 69 | 55 | 44 | 63 | 86 | 69 | 94 | 66 | 94 | 94 | 91 | 94 | 65 |
| sophia | 25 | 44 | 30 | 35 | 38 | - | 41 | 88 | 75 | 76 | 63 | 69 | 71 | 100 | 75 | 84 | 83 | 62 |
| bRHEAdBot | 4 | 7 | 24 | 44 | 49 | 69 | - | 51 | 64 | 79 | 59 | 65 | 83 | 99 | 81 | 96 | 98 | 61 |
| Ragnar | 40 | 50 | 32 | 26 | 50 | 13 | 46 | - | 44 | 71 | 63 | 69 | 73 | 88 | 81 | 73 | 85 | 56 |
| POLightRush | 0 | 25 | 29 | 38 | 31 | 44 | 34 | 38 | - | 71 | 69 | 69 | 73 | 100 | 75 | 91 | 100 | 55 |
| SaveTheBeesV4 | 14 | 9 | 21 | 43 | 31 | 59 | 38 | 47 | 66 | - | 50 | 57 | 81 | 86 | 85 | 90 | 93 | 54 |
| POWorkerRush | 13 | 13 | 21 | 29 | 31 | 44 | 44 | 56 | 38 | 89 | - | 75 | 49 | 94 | 81 | 81 | 96 | 53 |
| MyMicroRtsBot | 11 | 13 | 15 | 25 | 38 | 56 | 38 | 56 | 38 | 86 | 44 | - | 43 | 94 | 69 | 74 | 92 | 49 |
| NaiveMCTS | 0 | 11 | 17 | 22 | 34 | 27 | 15 | 26 | 29 | 69 | 56 | 58 | - | 92 | 46 | 60 | 84 | 40 |
| myBot | 1 | 6 | 21 | 20 | 39 | 48 | 28 | 41 | 43 | 77 | 39 | 40 | 50 | - | 55 | 66 | 66 | 40 |
| NIlSiBot | 0 | 13 | 18 | 18 | 31 | 25 | 13 | 13 | 31 | 63 | 31 | 38 | 51 | 81 | - | 58 | 73 | 35 |
| Predator | 1 | 7 | 13 | 6 | 12 | 21 | 11 | 16 | 14 | 56 | 22 | 28 | 44 | 73 | 43 | - | 45 | 26 |
| RandomBiasedAI | 0 | 1 | 15 | 0 | 4 | 15 | 6 | 9 | 4 | 52 | 4 | 13 | 18 | 85 | 39 | 39 | - | 19 |

Table 27: AnonymizedAI win rates split by competition job. Each map has 5 or 10 jobs that runs a round-robin tournament of all agents for 2 or 1 iterations, respectively. "Outlier" jobs are bolded. "Average" is the average win rate for all jobs. "Average Removing Outliers" is the average win rate for all jobs excluding "outlier" jobs.

| | basesWorkers8x8A | FourBasesWorkers8x8 | NoWhereToRun9x8 | basesWorkers16x16A | TwoBasesBarracks16x16 | DoubleGame24x24 | BWDistantResources32x32 | (4)BloodBath.scmB | Overall |
|---|---|---|---|---|---|---|---|---|---|
| | 98 | 97 | 97 | 100 | 94 | 99 | 49 | 34 | |
| | 100 | 94 | 97 | 100 | **4** | 96 | 53 | 38 | |
| | 100 | 95 | **33** | 100 | 92 | 100 | 54 | 38 | |
| | **5** | 92 | 94 | 98 | 94 | 97 | 58 | 39 | |
| | **28** | 95 | 100 | 100 | 94 | **9** | 58 | **19** | |
| | | | 100 | | | | | 39 | |
| | | | 97 | | | | | **19** | |
| | | | **30** | | | | | 44 | |
| | | | 97 | | | | | 36 | |
| | | | 95 | | | | | 36 | |
| Average | 66 | 95 | 84 | 100 | 75 | 80 | 54 | 34 | 74 |
| Average Removing Outliers | 99 | 95 | 97 | 100 | 93 | 98 | 54 | 38 | 84 |

## J  ADDITIONAL BEHAVIOR CLONING BENCHMARKS

Table 28: BC-Agent win rate in single player round-robin benchmark. Win rates above 50% are bolded.

| | POWorkerRush | POLightRush | CoacAI | Mayari | Overall |
|---|---|---|---|---|---|
| basesWorkers8x8A | **60** | **100** | **90** | 50 | 75 |
| FourBasesWorkers8x8 | **100** | **100** | **85** | **65** | 88 |
| NoWhereToRun9x8 | **100** | **100** | **83** | **55** | 85 |
| basesWorkers16x16A | 10 | **100** | **100** | 28 | 60 |
| TwoBasesBarracks16x16 | **100** | **100** | 43 | 20 | 66 |
| DoubleGame24x24 | 0 | **100** | **100** | 30 | 58 |
| BWDistantResources32x32 | 48 | **100** | **100** | **65** | 78 |
| (4)BloodBath.scmB | **100** | **63** | 20 | 40 | 56 |
| AI Average | **65** | **96** | **78** | 44 | 71 |

