# OpenReview forum: "A Competition Winning Deep Reinforcement Learning Agent in microRTS"
_ICLR.cc/2024/Conference — Submitted to ICLR 2024_

### Official Review · Reviewer_Pjfo · 2023-10-19

**Soundness:** 2 fair
**Presentation:** 2 fair
**Contribution:** 2 fair
**Rating:** 3
**Confidence:** 4

**Summary:**

This paper presents AnonymizedAI, the first Deep Reinforcement Learning (DRL) agent to win the IEEE microRTS competition. AnonymizedAI's training process involved transfer learning to specific maps, which was critical to its winning performance. The paper also discusses the challenges of debugging and fine-tuning a DRL implementation, as well as the potential benefits of combining Imitation Learning and DRL. The contributions of this paper are:
1. Introducing AnonymizedAI, the first DRL agent to win the IEEE microRTS competition.
2. Demonstrating the importance of transfer learning to specific maps in achieving competitive performance.
3. Providing insights into the challenges and potential benefits of using DRL in real-time strategy games.

**Strengths:**

Quality: The paper provides detailed information on AnonymizedAI's architecture, training process, and performance, as well as insights into the challenges and potential benefits of using DRL in real-time strategy games.
Clarity: The paper is well-organized and clearly written, with sections on Introduction, Related Work, Methodology, Results, and Conclusion.

**Weaknesses:**

the contribution lacks novelty, the network architecture mainly references existing algorithm networks. At the same time, a method that can well solve the performance under multiple different maps is not proposed.

**Questions:**

Q1: how to ensure the high performance of the model in the new hidden map or an untrained map?
Q2: the masking significantly reduces the action space per turn and makes training more efficient, and how to determine and obtain the action needs to be masked?

---

> ### Author Response · Authors · 2023-11-23
>
> **Generalization to different maps:** In Table 4, we show that BC-PPO-Agent performs as
> well as AnonymizedAI on the 3 maps AnonymizedAI needed fine-tuned models while only
> using map-size specific models. We think this is a promising direction to
> generalization, and we added a paragraph at the end of the Behavior Cloning Results
> Section stating this.
>
> **Generalization to larger maps:** We are working on generalizing to larger maps. We
> extended the Training on Larger Maps Discussion Section to include our current ideas.
>
> **What to action mask?:** We used MicroRTS-Py as the starting point and masked out a
> couple more cases where the action would cause nothing to happen. Implementing action
> masking is a major contribution of human knowledge to the learner but has been
> well-established as critical for learning in microRTS.

---

### Official Review · Reviewer_7eJX · 2023-10-21

**Soundness:** 4 excellent
**Presentation:** 3 good
**Contribution:** 2 fair
**Rating:** 5
**Confidence:** 4

**Summary:**

The authors describe their entry to the IEEE-CoG2023 microRTS competition, where their entry won, thus
becoming the first deep RL agent to win the competition. This is a competition to produce the best agent on the microRTS environment. RTS environments are difficult for RL agents because of their complex game structure with varying unit types and strategies, large action space, long episodes and sparse rewards.

The authors apply a number of implementation tricks to improve on existing methods, including using a value function that predicts three different rewards that vary throughout training and training 7 different networks and selecting based on the map. They then train an agent using behaviour cloning and then fine-tune it with deep RL.

**Strengths:**

The paper has a few notable strengths:
- The technical details of their implementation are very clearly described.
- Winning the IEEE MicroRTS competition while being the first deep RL agent to do so is clearly a notable achievement.

**Weaknesses:**

However, the paper has a few notable weaknesses. In particular, although an impressive feat of engineering, I gained little insight about which parts of their design were particularly important, how exploitable their method was, how to improve the performance on larger maps and the required deeper strategy and other important research questions surrounding designing a good RTS agent. The submission could be significantly improved by focussing more on understanding why and how the system itself works. For example, the paper could include ablations of the different design decisions, or attempt to train a deep RL agent on the largest map. Instead the paper is mostly a grab-bag of previously-known techniques that combine to produce a very good agent.

**Questions:**

- You mention that scaling the behaviour cloning loss by the number of units that could take an action was critical to get it to train. Have you investigated why this is?

---

> ### Author Response · Authors · 2023-11-23
> **Design decisions and ablation studies**
>
> We agree that the paper could be improved by
> a thorough analysis of design decisions. Unfortunately, we were time-constrained by the
> competition deadline and were experimenting with many different ideas going up to the
> deadline. We ran training runs with changes and kept the changes that made
> improvements or we believed would be useful down the line.
>
> One example of this is the scaling the policy loss by the number of units that could
> take an action, which was critical for our behavior cloning. We have now added our reasoning into the paper on
> why this is important; however, we believe this could be a candidate for an ablation
> study to determine if any learning rate works without this scaling.
>
> Other candidates for ablation studies include:
> - transfer learning smaller map agents to larger maps
> - varying reward weights, value loss coefficients, and entropy coefficients over time
> - cost-based rewards

---

### Official Review · Reviewer_nGmd · 2023-10-25

**Soundness:** 3 good
**Presentation:** 2 fair
**Contribution:** 3 good
**Rating:** 5
**Confidence:** 4

**Summary:**

This paper introduces a DRL model named AnonymizedAI, which is the first DRL method to win the IEEE microRTS competition. It defeated two prior competition winners in the IEEE microRTS (μRTS) competitions hosted at CIG and CoG. Its success largely benefits from iteratively fine-tuning the base policy and transfer learning to specific maps, which can be an economic method for training low-cost and efficient DRL agents.

**Strengths:**

**Originality:**

This paper presents a novel DRL training paradigm, which fine-tunes the base policy and transfers the policy to new scenarios. The proposed method won the IEEE microRTS competitions.

**Quality:**

To demonstrate the priority of the proposed method, this paper conducted massive experiments and presents detailed implementation of the novel method. The experimental results in the microRTS scenarios are convincing.

**Clarity:**

Overall, this paper is easy to follow. This article dedicates a considerable amount of text to elaborating on various technical details.

**Significance:**

This paper investigates an important research problem, i.e., training RTS AI efficiently with DRL. The proposed training method brings insights to the DRL community

**Weaknesses:**

Despite the fact that this paper presents a paper with convincing experimental results, I list some weaknesses:

1. This paper covers a considerable amount of text on technical details, such as policy networks and speeding up inference. However, it is hard to gain insights for training DRL agents on other complex scenarios, such as Mahjong and Stratego. [See the questions below]
2. The proposed method is a combination of prior methods. Contribution on DRL algorithm is limited.
3. Discussions on some related works are missing, such as SCC [1], it achieves top human performance defeating GrandMaster players in test matches and top professional players in a live StarCraft II event with order of magnitude less computation.

**Reference:**

[1] Wang et al. SCC: an efficient deep reinforcement learning agent mastering the game of StarCraft II.

**Questions:**

1. In Sec. 2.1, why self-play failed in UAS and GridNet?
2. Why did not you use supervised learning?
3. Why does AnonymizedAI load 7 different policy networks?
4. What is Squnet?
5. What is the key takeaway for readers who want to use this proposed pipeline to train DRL agents on other complex scenarios, such as Stratego, Mahjong, PUBG and Honour of Kings?

---

> ### Author Response · Authors · 2023-11-23
>
> **SCC and related works discussion:** Thanks for pointing us to the SCC paper. It's
> definitely impressive to accomplish similar results to AlphaStar with a smaller model
> and order of magnitude less compute. We knew of TStarBot-X as another successful
> StarCraft II agent trained with orders of magnitude less computation scale. We didn't go
> into these papers because these still required clusters of GPUs to train (SCC doesn't
> give specifics but order of magnitude less is tens of TPUs still). We thought microRTS and the
> Kaggle Lux competitions were more relevant given similar scale.
>
> Answers to Questions:
> 1. MicroRTS-Py had a bug where the second player would see resources (which should be
>   unowned) as owned by the first player. This was fixed for our agent. We moved that
>   mention up into the MicroRTS-Py section to make it more clear.
> 1. Behavior cloning treats a reinforcement learning problem as a supervised learning
>    problem by making the ground truth the actions taken by the agent being cloned.
> 2. AnonymizedAI trained 4 models fine-tuned to specific maps. 3 "general" models were
>    also trained for 3 different map size ranges (16x16 and smaller, up to 32x32, and up
>    to 64x64). Table 5 in the Competition Details Supplementary Section shows the 7
>    models' usage. The fine-tuned models significantly outperformed the general models on
>    their specific maps.
> 3.  squnet is the name of our fewer parameter model. It is short for "Squeeze U-Net"
>     because it uses squeeze excitation like DoubleCone while being shaped like a U-Net. It is not
>     a U-Net.
> 4. **Applicability to other strategy games:** We focused on
> microRTS because of its unique challenges and the existence of a benchmark without a
> strong DRL agent. We have used the same techniques to train
> agents for the Lux Season 2 competitions, which is a similar turn-based game requiring
> the coordination of many units individually. We have not tried other games, but we
> believe that many of these techniques would be applicable to other games. For example,
> creating a curriculum of simpler scenarios and gradually increasing difficulty or using
> a demonstration dataset to create a starting point for training are generally
> applicable.

---

### Official Review · Reviewer_AVMG · 2023-10-30

**Soundness:** 3 good
**Presentation:** 3 good
**Contribution:** 2 fair
**Rating:** 6
**Confidence:** 5

**Summary:**

The paper presents AnonymizedAI, a Deep Reinforcement Learning (DRL) agent, which is the first of its kind to secure a win in the IEEE microRTS competition. AnonymizedAI exploits a combination of carefully fine-tuned base policies and map-specific transfer learning to outperform previous competition winners. The paper describes the implementation of AnonymizedAI, including its 7 trained neural networks and the significant effort required to train and debug the agent, emphasizing the role of iterative fine-tuning and transfer learning in the agent's success. Furthermore, the authors discuss potential improvements in inference time and explore behavior cloning and its potential to bootstrap models with novel behaviors.

**Strengths:**

1. The major novelty of this work is the application of iterative fine-tuning and transfer learning to a DRL agent in the microRTS competition. The victory of AnonymizedAI in the competition demonstrates the effectiveness of the DRL approach in complex strategy games.

2. The paper is well-structured and clear in its explanations, making the complex mechanisms behind AnonymizedAI accessible to readers.

**Weaknesses:**

1. Although the developed agent provides an innovative solution in the realm of the µRTS competition, the novelty of the techniques employed—transfer learning and iterative fine-tuning—within the broader context of DRL is somewhat limited as these techniques are already widely adopted.
2. The authors didn't delve into an analysis of diverse self-play strategies that could potentially improve the agent's performance. Considerations for strategies beyond basic self-play, such as fictitious self-play[1] or more complex schemes[2], would have enriched this study.

- [1] Heinrich, Johannes, Marc Lanctot, and David Silver. "Fictitious self-play in extensive-form games." International conference on machine learning. PMLR, 2015.
- [2] Lin, Fanqi, et al. "TiZero: Mastering Multi-Agent Football with Curriculum Learning and Self-Play." arXiv preprint arXiv:2302.07515 (2023).

**Questions:**

1. It would be interesting to know whether the authors plan to explore the inclusion of more advanced self-play strategies, and how they believe these could potentially impact AnonymizedAI's performance.

2. The authors have mentioned the possibility of using the imitation learning approach for bootstrapping models with novel behaviors. Could they provide more insights into potential novel behaviors they are considering and how these could improve the agent's performance?

3. I did not find a description of the hardware information for training the agents in the paper, such as the CPU model and number of cores, GPU model and number of cores, the size of training memory usage, etc. I wonder if the author could provide this information.

---

> ### Author Response · Authors · 2023-11-23
> **Answers to Questions**
>
> 1. **Advanced self-play techniques:** In this paper we trained our agent against a
> combination of scripted opponents, the current agent, and past versions of the agent.
> Past versions of the agent were taken from periodic checkpoints during training, and the
> selected version was chosen randomly. As suggested, using fictitious self-play
> similar to TiZero or AlphaZero could be a promising direction for future work,
> especially for larger maps where multiple strategies are likely viable. A paragraph has
> been added to the Discussions sections.
> 1. **Novel behaviors from behavior cloning:** We see using the term "novel" was confusing.
> We replaced most instances of "novel" with "demonstrated", "competitive", and
> "effective".
> 1. **Training hardware details:** Sorry, this was an oversight. This has been added to
> Training Durations Supplementary Section.

---

### Author Response · Authors · 2023-11-23

We sincerely appreciate the feedback and insights provided by everyone. They are helpful in
identifying areas for improvement both in this paper and in ongoing work.

We've addressed reviewer-specific comments directly on the reviews. Additionally, we
want to address the following generally:

**Novelty to DRL field:** We agree with the interpretation that AnonymousAI uses many
best-practices across DRL. We believe the novelty of this paper is specifically in
microRTS and environments where many units have to be individually controlled. This is
an area where hand-scripted AI agents have been dominant, as evidenced by prior winners
of microRTS competitions and the latest Lux Season 2 competitions on Kaggle. This paper
represents a milestone for DRL in this domain on single GPU hardware, which is more
common in an academic setting.

Minor comments:
- We forgot to add the Behavior Cloning schedule for 16x16 maps. This has been added to the
  Behavior Cloning Supplementary Section.

---

### Meta-Review · Area_Chair_RZCC · 2023-12-14

**Metareview:**

I feel somewhat bad not recommending this paper for publication at ICLR. It won a tough, contested competition, and I think it therefore has a strong claim to be getting published. However, reviewers are not very enthusiastic about it, even somewhat negative. Of course, I could override them, but I see where they're coming from. While the paper is technically impressive, it comes down to the question of what we actually learn form it. This is one of those papers that include many small advances instead of one big advance. As a result, it comes out somewhat unfocused. And I guess the nature of CS conferences is such that it prioritizes "one big advance" (one weird trick?) papers. So, I'm sorry. Better luck at another venue, perhaps with a more focused paper.

**Justification For Why Not Higher Score:**

Reviewers don't like it.

**Justification For Why Not Lower Score:**

N/A

---

### Decision · Program_Chairs · 2024-01-16

Reject